# Adhesive Capsulitis of the Shoulder: Current Concepts on the Diagnostic Work-Up and Evidence-Based Protocol for Radiological Evaluation

**DOI:** 10.3390/diagnostics13223410

**Published:** 2023-11-09

**Authors:** Riccardo Picasso, Federico Pistoia, Federico Zaottini, Giovanni Marcenaro, Maribel Miguel-Pérez, Alberto Stefano Tagliafico, Carlo Martinoli

**Affiliations:** 1IRCCS Ospedale Policlinico San Martino, Largo Rosanna Benzi 10, 16145 Genova, Liguria, Italy; riccardo.picasso@hsanmartino.it (R.P.); federico.zaottini@hsanmartino.it (F.Z.); alberto.tagliafico@unige.it (A.S.T.); carlo.martinoli@unige.it (C.M.); 2Department of Health Sciences (DISSAL), Università di Genova, Via Antonio Pastore 1, 16132 Genova, Liguria, Italy; 4342889@studenti.unige.it; 3Unidad de Anatomía y Embriología Humana, Departamento de Patología y Terapéutica Experimental, Facultad de Medicina y Ciencias de la Salud (Campus de Bellvitge), Universitat de Barcelona, 08904 Barcelona, Spain; mabelmig@gmail.com

**Keywords:** adhesive capsulitis, frozen shoulder, MRI, MRA, ultrasound

## Abstract

Adhesive capsulitis is an idiopathic and disabling disorder characterized by intense shoulder pain and progressive limitation of active and passive glenohumeral joint range of motion. Although adhesive capsulitis has been traditionally considered a diagnosis of exclusion that can be established based on a suggestive medical history and the detection of supporting findings at the physical exam, imaging studies are commonly requested to confirm the diagnostic suspicion and to exclude other causes of shoulder pain. Indeed, clinical findings may be rather unspecific, and may overlap with diseases like calcific tendinitis, rotator cuff pathology, acromioclavicular or glenohumeral arthropathy, autoimmune disorders, and subacromial/subdeltoid bursitis. Magnetic resonance imaging, magnetic resonance arthrography, and high-resolution ultrasound have shown high sensitivity and accuracy in diagnosing adhesive capsulitis through the demonstration of specific pathological findings, including thickening of the joint capsule and of the coracohumeral ligament, fibrosis of the subcoracoid fat triangle, and extravasation of gadolinium outside the joint recesses. This narrative review provides an updated analysis of the current concepts on the role of imaging modalities in patients with adhesive capsulitis, with the final aim of proposing an evidence-based imaging protocol for the radiological evaluation of this condition.

## 1. Introduction

Adhesive capsulitis (AC), also known as frozen shoulder, is an idiopathic and disabling disorder characterized by intense pain and the progressive limitation of the active and passive glenohumeral range of motion. This condition was first described in 1872 by Duplay under the name scapulohumeral periarthritis, whereas the terms frozen shoulder and AC were introduced later by Codman and Neviaser in 1934 and 1945 [1,2,3]. Although AC has been traditionally considered a diagnosis of exclusion that can be established based on a suggestive medical history and the detection of supporting findings at the physical exam, imaging studies are commonly requested in clinical practice to confirm the diagnostic suspicion and to exclude other causes of shoulder pain [4]. Magnetic resonance imaging (MRI) and magnetic resonance arthrography (MRA) have shown great accuracy in diagnosing AC through the demonstration of specific abnormalities involving the glenohumeral joint capsule and the pericapsular soft tissues and, at present, are considered the gold standard for the radiological evaluation of these patients [5,6]. On the contrary, the use of high-resolution ultrasound (US) is not supported with consensus-based guidelines from the European Society of Musculoskeletal Radiology [7] or the European Society for Ultrasound in Medicine and Biology (EFSUMB) [8,9], despite several research studies, including a recent meta-analysis, evidencing comparable levels of accuracy of US and MRI in diagnosing AC [10,11,12]. The aim of this work is to provide a critical review of recent concepts on the role of imaging modalities in AC and to propose a reasoned and evidence-based protocol for the radiological evaluation of this disorder.

## 2. Relevant Anatomy

The glenohumeral joint is a ball and socket joint between the scapular glenoid and the humeral head. The shallow surface of the glenoid cavity and its relatively small dimension respective to the humeral head account for the wide range of motion of the shoulder, but at the same time make this latter intrinsically unstable. A loose and lax capsule envelops the joint and permits the large mobility of the articular ends [13] (Figure 1). The capsule inserts medially on the circumference of the bony glenoid and laterally below the anatomical neck of the humerus. It is redundant on its inferior side where it forms a wide pocket, named the axillary pouch, which constitutes the major recess of the joint cavity. The thickness of the glenohumeral joint capsule is not homogeneous; the width of its insertions on the articular ends varies within the different joint quadrants. Anatomic dissection and quantitative analysis through micro-computed tomography demonstrated that the capsular insertion is wider in the caudal part of the humeral neck than in the cranial, whereas the capsule is thicker in its anterior and inferior portions. The thinness of the superior and posterior capsule, as well as its narrower bone insertion, are supposed to be compensated by the presence of the overlying rotator cuff tendons, which provide joint stability in those areas [14].

Moreover, specific zones of the capsule are reinforced by extrinsic and intrinsic ligaments that contribute to the stabilization of the shoulder. The superior, middle, and inferior glenohumeral ligaments are rather inconstant and highly variable thickenings of the capsule that are best observed from the inner side of the joint cavity during arthroscopic procedures [15]. There is an ongoing debate about the exact anatomy of the glenohumeral ligaments, which is, however, beyond the scope of this work. Generally speaking, the superior glenohumeral ligament is the most constant and is composed of a direct and an oblique group of fibers, arising from the anterior glenoid labrum and the supraglenoid tubercle, respectively [16]. The direct bundle runs parallel and anterior to the long head of the biceps tendon toward the lesser tuberosity. There it inserts together with the medial arm of the coracohumeral ligament and the superior sling of the subscapularis tendon to form the biceps pulley, which plays a role in the stabilization of the intra articular portion of the long head of the biceps tendon. The oblique bundle crosses over the long head of the biceps tendon and fuses with the lateral arm of the coracohumeral ligament. The middle glenohumeral ligament is the most inconstant, as it has been described in only 60–85% of people, and may present significant variations in thickness, shape, and insertions. In its most typical appearance, the middle glenohumeral ligament originates from the supraglenoid tubercle in close contiguity to the superior glenohumeral ligament and inserts in the humeral neck just medial to the lesser tuberosity or into the deep surface of the subscapularis tendon [17]. The inferior glenohumeral ligament has been reported in between 72% and 93% of subjects [15,18]. When present, the ligament has a hammock-like shape and is constituted by a constant anterior band arising from the anteroinferior aspect of the glenoid labrum between 3 and 5 o’clock and an inconstant posterior band, which can be observed in 41–73% of cases and arises from the posteroinferior aspect of the glenoid labrum between 7 and 9 o’clock. In between the two bands, the axillary pouch represents the most dependent portion of the glenohumeral joint capsule. However, the exact anatomy of the inferior glenohumeral ligament is debated, with other authors indicating that it can be found in every subject as focal thickenings of the inner layer of the anteroinferior and posteroinferior joint capsule [19]. Overall, the thickness of the inferior glenohumeral ligament decreases from anterior to posterior, averaging 2.8 mm in the most anterior part and 1.7 mm in the posterior portion, even if many discrepancies are found in anatomic descriptions [20].

The coracohumeral ligament is a complex and relevant stabilizer of the glenohumeral joint that originates from the outer margin of the coracoid and has multiple distal insertions into the greater and lesser tuberosity, the joint capsule, the supraspinatus tendon, and the subscapularis [21]. More in detail, after detaching from the coracoid as a single bundle, the coracohumeral ligament diverges in a medial and a lateral arm at the level of the rotator interval. The superficial fibers of the lateral arm cover the anterior portion of the supraspinatus tendons and fan out laterally and posteriorly, merging with the periosteum of the greater tuberosity [22]. The deep fibers of the lateral arms travel from medial to lateral and from anterior to posterior underneath the supraspinatus tendons, merging with the deep fibers of this latter, part of the superior glenohumeral ligament, and the glenohumeral joint capsule to form the so-called superior complex and the rotator cable [23,24]. The medial arm envelop the cranial part of the subscapularis muscle and tendon and insert into the lesser tuberosity at the level of the rotator interval, merging with the superior glenohumeral ligament and the cranial slip of the subscapularis tendon to form the reflection pulley [25]. The coracohumeral ligament plays a critical role in stabilizing the glenohumeral joint in the upright position, limiting inferior subluxation of the humeral head [26]. Fibrosis, hyalinization, and fibrinoid degeneration of the coracohumeral ligament have been reported among the major histological findings in AC [27].

## 3. Epidemiology and Risk Factors

Epidemiological data show that AC has a prevalence between 2% and 5% of the general population. In 70% of cases it involves females, and usually affects people in the 5th to 7th decades of life [28,29]. Although in most instances the onset of AC does not follow any specific event and the disease is considered idiopathic, in a minority of cases it may be secondary to surgical procedures or locoregional trauma. In addition, a genetic predisposition has been evidenced in patients with a positive family history and expression of HLA-B27 [30]. Several risk factors have been linked to the development of AC, including diabetes, cerebrovascular accident, thyroid disease, autoimmune disorders, coronary artery disease, and Dupuytren’s syndrome [29]. In particular, it is worth mentioning the strong association between AC, diabetes, and cerebrovascular diseases, with a prevalence of AC ramping up to 10.3% and 22.4% in patients with type I and type II diabetes, and 25.3% in people who had a stroke within the previous six months [31,32]. Finally, the inflammatory process involving the joint capsule and the consequent immobilization of the shoulder have been postulated as a potential trigger of AC in patients with calcific tendinitis, but the pathogenic relation between the two conditions has not been completely elucidated [33,34,35].

## 4. Pathogenesis

Although the pathogenesis of AC has yet to be elucidated, recent evidence points out that this condition is the result of both inflammatory and fibrotic processes. Macroscopically, the hallmarks of AC include shrinking and loss of the synovial layer of the capsule, inflammation, edema, and thickening of the rotator interval and the coracohumeral ligament, decreased capsular volume, and adhesions of the axillary recess walls to themselves and to the humerus [36]. Histological sampling demonstrated a fibrotic process characterized by fibroblasts immersed in a matrix of type I and type III collagen. The observation of myofibroblast in specimens has been linked to the development of capsular contracture. A dysregulation in the metabolism of collagen is also hypothesized, due to the anomalous expression of metalloproteinases and tissue inhibitors of metalloproteinases in affected tissues [37,38]. The existence of an inflammatory process underpinning the fibrotic response is suggested by the recent demonstration of inflammatory cytokines, neoangiogenesis, and neoinnervation in capsular and bursal samplings [39]. Of note, the overexpression of intercellular adhesion molecule-1 (ICAM-1), which is a transmembrane protein involved in the inflammatory response, has been observed in both patients with AC and diabetes, possibly explaining the common association between the two conditions [40].

## 5. Natural History

The natural history of AC has been traditionally described following three or four clinical stages, which correspond to specific histologic alterations that are observable in the joint capsule of affected patients [41,42].

First stage (painful stage): It occurs in the first three months, and it is characterized by the insurgence of ill-defined ache around the deltoid area, which tends to worsen at night. Mild restriction of glenohumeral joint range of motion may also be observed. From a histological point of view, this phase is distinguished by the development of an inflammatory infiltrate, synovitis, and hypervascularity at the level of the glenohumeral joint capsule.Second stage (freezing stage): Between the third and the ninth month the patient experiences progressive joint stiffness and the increase of pain. Joint stiffness is particularly evident in forward flexion, abduction, and extra rotation. The capsule shows macroscopic alteration with thickening and hypervascularity of the synovial membrane, disorganized deposition of collagen, and adhesions. The first and second stages have been grouped together in recent descriptions.Third stage (frozen stage): This phase may last until the fourteenth month and is distinguished by severe restriction of joint movements and an initial decrease of pain, which in this phase is less evident at rest but remains intense during passive mobilization of the glenohumeral joint.The fourth stage (thawing stage): The last phase is distinguished by the spontaneous resolution of joint stiffness and pain, which may require up to two years. Histologically, mature and adhering hypercellular collagen is detected in the capsular tissue in the third and fourth stages whereas inflammatory signs are less evident.

The self-recovery of all subjects with AC has been questioned in recent prospective studies, in which most of the spontaneous improvements of range of motion were observed in the early phases, whereas patients with persistent symptoms over a long period of time demonstrated a scarce tendency to ameliorate without treatment [43]. Overall, recent studies point out that up to 40% of patients still complain of restricted shoulder motion four years after the onset of the disease [44]. These data also suggest that early diagnosis and treatment may be critical in decreasing disease burden and reducing time to recovery.

## 6. Clinical Diagnostic Work-Up

AC has been traditionally considered a clinical diagnosis, based on the detection of glenohumeral joint stiffness and intense shoulder pain lasting for more than four weeks [45]. From a clinical point of view, pain is described as dull and poorly localized, with possible radiation in the area of the long head of the bicep tendon. Whereas in the first phases the pain is usually present at rest and has typical overnight exacerbation, in the later stages it is principally evoked by joint movements. At the clinical exam, both active and passive ranges of motion are affected, and joint stiffness is worst in extra rotation, abduction, and forward flexion. In advanced cases, joint stiffness may be so important as to reduce arm swinging during gait. Although laboratory tests are supposed to yield normal results, they may be requested to exclude commonly associated conditions such as hyperthyroidism, autoimmune disorders, and diabetes. Overall, clinical findings may be rather unspecific and may overlap with diseases like calcific tendinitis, rotator cuff pathology, acromioclavicular or glenohumeral arthropathy, autoimmune disorders, and bursitis. Indeed, the low accuracy and reproducibility of physicians in diagnosing shoulder disorders based only on clinical examination has been repeatedly demonstrated in recent works [46,47]. Moreover, a Delphi consensus between rehabilitation and physical medicine specialists, orthopedic surgeons, physical therapists, chiropractors, and osteopaths failed in establishing a set of definite criteria for the clinical diagnosis of AC, concluding that findings at the medical exam may be only suggestive for this condition [48]. In conclusion, the absence of specific tests and the broad differential diagnosis make AC impossible to diagnose on clinical grounds with a sufficient degree of accuracy. Even if it is a matter of debate, delays in diagnosis and suboptimal treatment are likely to impact prognosis and time to recovery [49].

## 7. Role of Imaging

To date, imaging has a controversial role in the diagnostic workup of AC. An explanatory case is provided by a highly cited work supporting the clinical diagnosis of frozen shoulder that included in the diagnostic criteria for this condition a plain film negative for calcific tendonitis and osteoarthritis [45]. Despite the lack of a strong consensus, imaging studies are generally referred to as supporting but not necessary in the diagnostic workup and their role has been limited to atypical or refractory cases [50]. On the other hand, a huge number of papers explored the potential of imaging modalities in diagnosing AC and several radiological findings have been demonstrated to be highly sensitive and specific for this condition. Even if in the past conventional arthrography played a significant role in the diagnosis of AC through the demonstration of indirect findings such as reduced capsular distension and extravasation of contrast agent [51], the ability of MRI and US in directly demonstrating the thickening of the glenohumeral capsule and in disclosing typical pathological alterations has made these modalities the most employed in patients with AC. Furthermore, the possibility to use US to target percutaneous therapies (e.g., hydrodistension, drug injection) makes this technique increasingly appealing for the diagnostic and therapeutic management of AC [52].

In the next two paragraphs, the most recent evidence about the diagnostic performance of MRI and US are discussed, and the main findings are summarized in Table 1 and Table 2, respectively.

## 8. Magnetic Resonance Imaging

Due to its superb contrast resolution and its ability to demonstrate the whole glenohumeral capsule and pericapsular soft tissue, MRI is a commonly used diagnostic tool for the evaluation of patients with suspected AC and has been referred to as the gold standard amongst imaging modalities in this context (Figure 2). A recent meta-analysis identified six main MRI findings (i.e., the coracohumeral ligament thickening, the fat obliteration of the rotator interval, the inferior glenohumeral ligament hyperintensity and thickening, and contrast enhancement of the axillary joint capsule and the rotator interval) as the most accurate for AC diagnosis [53] (Figure 3). However, due to the unavailability of raw data, the authors were not able to provide precise cut-off values for any single parameter.

The thickening of the rotator interval and coracohumeral ligament are considered specific signs, although poorly sensitive, of AC. Various cut-off points were proposed in the past, and the optimal value is still a matter of debate. Mengiardi et al. found that a rotator interval capsule thickness ≥7 mm had a specificity of 86% but only a sensitivity of 64% [54]. Similarly, they reported that a coracohumeral ligament thickness ≥4 mm had high specificity (95%) but lower sensitivity (59%). In another study by Jung et al., it was suggested that a rotator interval thickening of over 6 mm on sagittal oblique proton-density images may correlate with the patient’s range of rotational motion [55]. Synovial proliferation around the rotator interval can be observed as capsule thickening with intermediate to low T1 signal intensity, hyperintense signal on fluid-sensitive sequences, and enhancement after contrast administration. While the synovial obliteration of the triangular fat pad inferior to the coracohumeral ligament has been identified as a sign with high specificity (100%), its sensitivity remains poor (32%) [54]. Referred to as the “subcoracoid triangle sign”, the obliteration of the subcoracoid fat pad has been more frequently observed in early clinical stages 1 and 2 of AC [56]. The visibility of this sign is optimal on sagittal oblique images and can be easily evaluated using conventional MRI techniques, where it appears hypointense relative to subcutaneous fat on T1-weighted images.

Several research studies have shown that hyperintensity and thickening of the inferior shoulder capsule are indicative of AC. In particular, hyperintensity in the axillary pouch/inferior glenohumeral ligament complex on MRI using non-arthrography T2-weighted fat-suppressed sequences demonstrated high sensitivity (85.3–88.2%) and specificity (88.2%), and low variability among different observers with a kappa value of 0.85 [57]. Regarding inferior capsule thickness, a first MRI study with a limited number of participants evidenced that when the measurements of the joint capsule in the axillary recess exceed 4 mm on T1 oblique coronal MR images, it suggests the diagnosis of AC with a sensitivity of 70% and a specificity of 95% [58]. In Jung et al.’s study, performed with conventional MRI, a threshold value of 4.5 mm in axillary recess capsule thickness measured on T1 oblique coronal images demonstrated the greatest diagnostic accuracy for AC, with a sensitivity, specificity, and overall accuracy of 91%, 90%, and 90%, respectively [55]. Other studies have shown that the thickness of the axillary recess is related to the clinical stage. Sofka et al. found a mean axillary pouch thickness of 7.5 mm for stage 2, also demonstrating a statistically significant correlation between the hyperintense signal of the capsule on proton density sequences obtained with conventional MRI and this clinical stage [56]. Contrary to the claim that MRA may have some advantages in measuring inferior capsular thickness, it is important to consider potential risks and complications associated with this technique, including hemorrhage and septic arthritis. Moreover, the sensitivity and specificity of inferior glenohumeral thickening on MRI were not significantly different from those on direct MRA found in a recent meta-analysis [52]. Overall, despite the potential of MRA in disclosing ancillary findings, such as leakage of contrast agent anterior to the medial margin of the scapula, reduced distension of the axillary recess, pseudo-synovitis over the cranial border of the subscapularis tendon and the biceps anchor, and widening of the subscapular recess [1] (Figure 4), the intra-articular injection of contrast agents appears unjustified in patients with AC.

Similarly, the utility of intravenous gadolinium in evaluating joint capsule vascularization and thickening is a topic of debate. Ahn et al. found a statistically significant positive linear correlation between the grade of axillary recess capsule enhancement, the thickness of the joint capsule, and pain intensity in individuals diagnosed with AC; however, no association was observed between the severity of limitation in forward elevation, external or internal rotation and the degree of axillary recess capsule enhancement [59]. Moreover, in another study, no substantial effects on the diagnosis of AC emerged between conventional MRI and gadolinium-enhanced MRI, despite the intravenous administration of contrast agent appearing to have some effect in increasing the reader’s confidence in measuring the joint capsule [60].

In conclusion, although MRA and MRI with intravenous contrast injection may provide additional information about the status of the capsule and may increase the reader’s confidence in diagnosing AC, conventional MRI without contrast administration has been proven to be accurate enough for diagnosis and should be preferred considering its lower invasiveness and cost [53,60].

## 9. Ultrasound

The progressive refinement of US technology and the amelioration of high-frequency transducers have enabled a precise evaluation of the glenohumeral joint capsule and the pericapsular ligaments, which now can be identified and discriminated from the surrounding soft tissues (Figure 5). Several research studies pointed out the great accuracy of US in diagnosing AC, and a recent meta-analysis demonstrated a combined sensitivity of 88% (95%CI: 74–95) and specificity of 96% (95%CI: 88–99) when inferior capsule and coracohumeral thickening, rotator interval abnormality, and restricted range of motion are evaluated with this modality [5,10,11] (Figure 6). According to the study of Michelin et al., the mean thickness of the axillary pouch in patients with AC measured with US was 4 mm versus 1.3 mm in asymptomatic controls [61]. A later study by Kim confirmed the ability of US to disclose the pathological thickening of the capsule at the axillary recess in patients with unilateral AC. The authors measured the capsule at its widest portion, including both the humeral and glenoid layer side and evidenced a mean value of 4.4 mm for the affected shoulder and 2.2 mm for the unaffected shoulder (*p* < 0.001) [62]. In the same study, the capsular thickness measured with US showed a correlation with the MRI measurements (*p* < 0.001, r = 0.83). Similarly, other authors suggested that a cutoff value of 4 mm for the axillary pouch yielded a sensitivity of 93.8% and a specificity of 98% in diagnosing AC and that a difference of 60% between the affected and the unaffected side may help in disclosing this condition also in patients with suggestive symptoms but axillary recess thickness less than 4 mm [63].

Regarding the coracohumeral ligament mean thickness in healthy subjects and patients with AC, a lack of consensus can be found in different US studies, intuitively depending on the scanning technique adopted for the identification and measurement of this structure. Homsi et al. evaluated the coracohumeral ligament with US by measuring its thickness in both long- and short-axis cross-sections and considering the maximal thickness value obtained. Using this method, an average coracohumeral ligament thickness of 3 mm was demonstrated in shoulders affected by AC. In comparison, painful shoulders exhibited an average coracohumeral ligament thickness of 1.39 mm, whereas asymptomatic shoulders had an average coracohumeral ligament thickness of 1.34 mm [64]. In a study by Tandon et al., the thickness of the coracohumeral ligament was measured in the oblique axial plane while keeping the arm externally rotated. Even with this method, patients with AC exhibited a significantly higher coracohumeral ligament thickness (1.2 mm) compared to both the group with painful shoulders (0.54 mm) and the control group (0.4 mm) [65]. In patients with AC, Lee et al. found a higher prevalence of distension of the long head of biceps tendon sheath in the affected shoulder respect to the contralateral side while Park et al. demonstrated a greater amount of effusion within the long head biceps tendon sheath in patients with AC compared to patients with other causes of painful shoulder [66,67]. Moreover, they demonstrated a negative correlation between the amount of effusion within the long head of the biceps tendon sheath and the glenohumeral range of motion. However, it has to be considered that any condition determining inflammation of the joint capsule may lead to fluid distension of the long head of the biceps tendon sheath and consequently this finding cannot be considered specific for AC.

A study that focused on dynamic US evidenced the presence of the subacromial gliding limitation of the supraspinatus tendon in 70.1% of patients with AC [68]. Additionally, the limitation of the supraspinatus tendon gliding beneath the acromion was demonstrated to be inversely correlated with the maximum amount of intra-articular injection for MRA. This correlation can probably be attributed to the reduced joint capacity resulting from increased capsular stiffness. Regarding rotator interval abnormalities, Lee et al. analyzed 30 patients with AC to investigate the correlation between local hypervascularity observed using color Doppler US and arthroscopic findings. A significant majority (86.67%) of these patients displayed color Doppler signals and hypoechoic echotexture in the rotator interval, which were correlated to the fibrovascular inflammatory soft tissue changes noted during arthroscopy [69]. US evidence of hypervascular soft tissue within the rotator interval demonstrated high sensitivity (97%) and specificity (100%) for the diagnosis of AC.

Other authors investigated the potential of novel Doppler techniques enabling the analysis of microflows in detecting hypervascularization of the subcoracoid triangle in patients with AC, comparing the diagnostic efficacy of microflow analysis with the one of gray-scale findings and standard Doppler techniques. In their work, microvascular analysis demonstrated a high performance in diagnosing AC, reaching a sensitivity of 76.92% and a specificity of 91.43% by setting the cutoff of the maximal area of vascular flow detected during the examination of the subcoracoid triangle at 1.31 mm^2^ [70]. Moreover, microvascular analysis outperformed power Doppler in disclosing abnormal vascularization in the subcoracoid fat triangle and demonstrated a statistically significant correlation with the limitation of glenohumeral range of motion. Finally, the use of US arthrography in AC was investigated in a recent paper by Cheng by injecting a sonographic contrast agent into the glenohumeral joint [71]. The injection fluid volume did not differ significantly between healthy controls and patients (19.0 ± 0.22 mL vs. 18.3 ± 0.29 mL, *p* = 0.07). However, patients with AC had a significantly smaller axillary recess volume compared to control subjects (1.14 ± 0.13 mL vs. 1.59 ± 0.08 mL, *p* < 0.01). Additionally, patients with AC showed more frequent filling defects in the joint cavity and synovitis-like abnormalities than control subjects (91.1% vs. 13.3% and 75.6% vs. 22.2%, respectively).

## 10. Suggested Imaging Protocol for Patients with Clinically Suspected AC

The proposed protocol is summarized in Figure 7. As exposed in the previous paragraphs, recent evidence points out that US is a valuable diagnostic tool for patients with AC and it is able to detect most of the anomalies evident on MRI. Furthermore, US offers significant advantages such as short examination time, low costs, and great accessibility, allows a fast comparative evaluation of the affected and the unaffected shoulder, and can be performed in patients with MRI non-compatible implants. In addition, US may effectively help in the differential diagnosis, disclosing signs of calcific tendinitis, rotator cuff pathology, and glenohumeral/acromioclavicular arthropathy [7,8], thus replacing conventional radiography in this task. Even if there is a lack of definite data on the cost-effectiveness of this modality in AC, recent evidence suggests that early diagnosis and treatment may impact patients’ prognosis [43]. Based on these considerations, US evaluation may be appropriate in patients with clinically suspected AC, to confirm the diagnosis and exclude other causes of shoulder pain. The exam should start by measuring the glenohumeral joint thickness at the axillary recess using a cut-off value of 4 mm. In cases where the capsular thickness at the axillary recess is inferior to 4 mm, the contralateral capsule should be evaluated to disclose any differences between the affected and the unaffected side. A difference of more than 60% may be considered a suggestive sign of adhesive capsulitis, even if further studies are needed to confirm this cutoff. In our experience the axillary recess capsule thickness can be measured even in those patients with limited range of motion, asking them to lay down on the examination bed and gently helping them to abduct the shoulder. Considering the lack of agreement on specific cutoff values for the rotator interval and the coracohumeral ligament thickness, these structures should be evaluated on both sides to disclose any difference in their shape and thickness, which in the proper clinical setting can be considered suggestive of AC, especially when the evaluation of the inferior capsule is inconclusive. Effusion in the long head of the biceps tendon sheath should be considered as supporting for an AC diagnosis only when associated with other suggestive findings, as this signs lacks specificity and may be found in several other conditions. The detection of synovitis or hypervascularization in the rotator interval and the subcoracoid triangle should be regarded as supporting but not necessary findings. On the other hand, MRI should be considered in doubtful cases after US with the aim of disclosing potential differentials that are not completely amenable to this latter modality, such as glenohumeral osteoarthritis, labral tears, and bone fracture or edema. In adhesive capsulitis, T2-hyperintensity and thickness superior to 4–4.5 mm of the axillary recess capsule and obliteration of the subcoracoid fat triangle should be considered diagnostic. Thickening of the coracohumeral ligament should also be evaluated, even if with this modality the agreement on a cutoff value is also poor. The use of MRAs and contrast-enhanced MRIs appear unjustified in patients with clinically suspected AC.

## 11. Conclusions

Technological advancements have greatly increased the diagnostic power of US and MRI, and an increasing number of research studies are confirming the potential of these modalities in disclosing sensitive and specific signs of AC. Considering the intrinsic limitations of the clinical exam and the low cost, high availability, and high accuracy of US, this latter modality may be recommended in every patient with clinically suspected AC with the aim of confirming the diagnosis and excluding other causes of shoulder pain. On the other hand, the high cost and lower availability of MRI make the use of this technique suitable only in cases where US and clinical findings do not reach a definite diagnosis. However, further studies are needed to assess the cost-effectiveness of the imaging evaluation of patients with suspected AC. In particular, imaging and clinical diagnostic protocols should be compared in term of time elapsing from symptoms onset and start of the appropriate treatment, disabling symptoms duration, and overall economic burden of the diagnostic-therapeutic work-up.

## Figures and Tables

**Figure 1 diagnostics-13-03410-f001:**
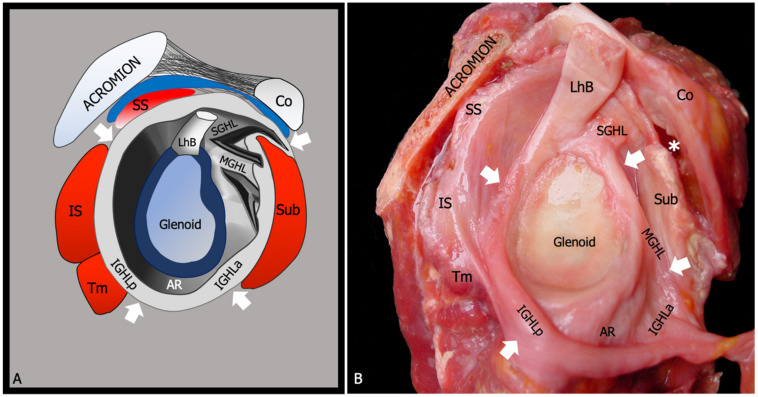
Glenohumeral joint capsule. (**A**) Schematic drawing and (**B**) anatomic dissection show the glenohumeral joint capsule (arrows) and its relationship with the overlaying rotator cuff tendons. The superior glenohumeral ligament (SGHL), the middle glenohumeral ligament (MGHL), and the anterior (IGHLa) and the posterior (IGHLp) bands of the inferior glenohumeral ligament are demonstrated as focal thickening and folding of the anteroinferior capsule. SS, supraspinatus tendon; IS, infraspinatus tendon; Tm, teres minor tendon; Sub, subscapularis tendon; LhB, long head of the biceps tendon. Co, coracoid process; AR, axillary recess; asterisk, subscapularis recess.

**Figure 2 diagnostics-13-03410-f002:**
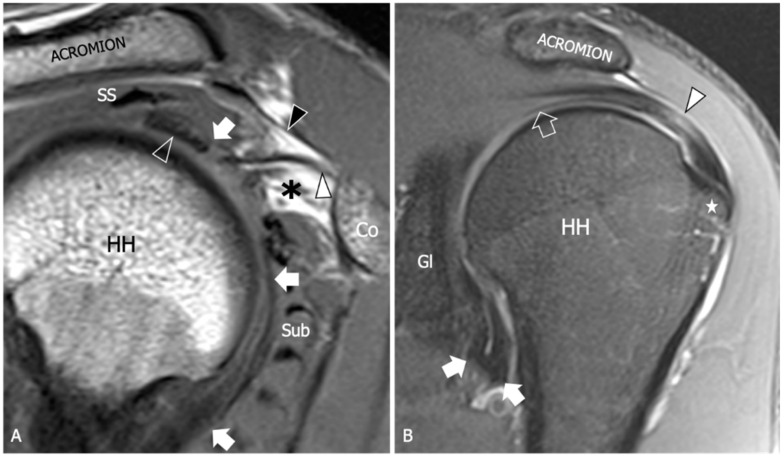
Glenohumeral joint capsule, normal MRI findings. (**A**) Sagittal tSE T1-weighted MRI image shows the joint capsule (arrow) as a structure of intermediate signal located deep to the rotator cuff tendons. Note the intra-articular part of the long head of the biceps tendon (outlined arrowhead) running on the inner surface of the capsule. The coracohumeral ligament (white arrowhead) is demonstrated as a thin and low signal fibrillar structure running from the coracoid (Co) to the humeral head (HH). The subcoracoid fat triangle (asterisk) is a fat-filled space delimited by the coracohumeral ligament, the joint capsule, and the subscapularis (Sub) muscle. SS, supraspinatus muscle; black arrowhead, coracoacromial ligament. (**B**) Coronal tSE fat-suppressed T2-weighted MRI scan shows the inferior part of the joint capsule (arrow) as a low signal folding that delimits the axillary recess. Note the thin superior capsule (outlined arrow) located underneath the supraspinatus muscle and tendon (arrowhead). Gl, scapular glenoid; HH, humeral head; star, greater tuberosity.

**Figure 3 diagnostics-13-03410-f003:**
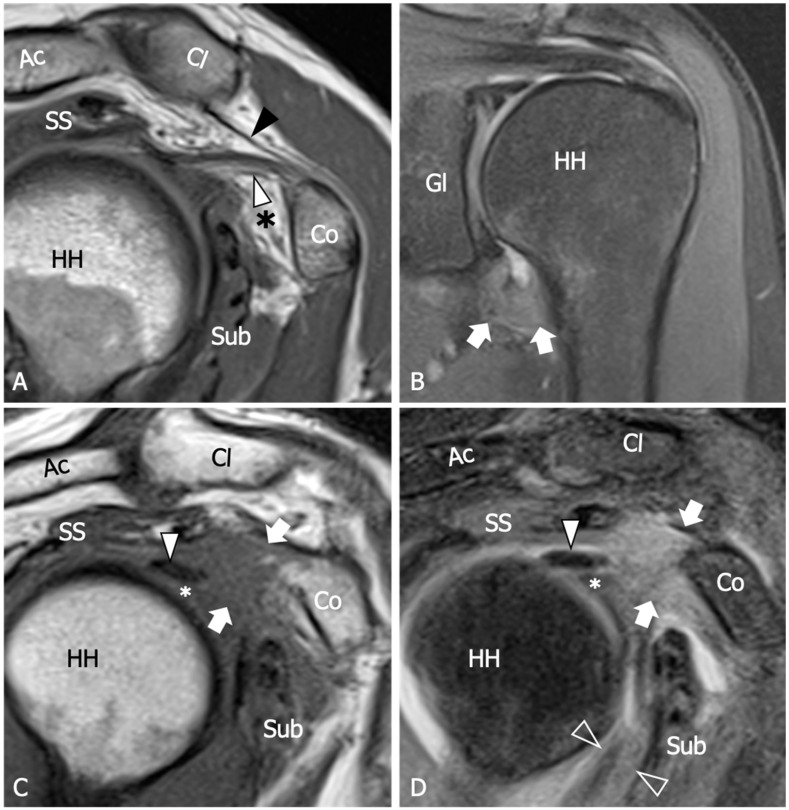
Adhesive capsulitis, spectrum of MRI findings. (**A**) Sagittal tSE T1-weighted MRI scan from a 42 year old woman with a three-month history of shoulder pain demonstrates mild thickening of the coracohumeral ligament (white arrowhead) and initial effacement of the subcoracoid fat triangle (asterisk) by hypointense synovium. Black arrowhead, coracoacromial ligament. (**B**) Coronal tSE Proton Density MRI scan from a 68 year old woman with recent onset of pain and progressive limitation of glenohumeral ROM shows a marked thickening of the inferior capsule (arrows), which appears edematous and demonstrates increased signal intensity in fluid-sensitive sequences. (**C**) Sagittal tSE T1-weighted and (**D**) Sagittal tSE fat-suppressed T2-weighted MRI images from a 70 year old male with a one-year history of severe limitation of active and passive shoulder motion demonstrate complete obliteration of the subcoracoid fat triangle by synovial tissue (arrows), which is also extended underneath the long head of the biceps tendon (arrowhead) in the area of the pulley (asterisk). The coracohumeral ligament appears embedded by the synovium. Note severe thickening and hyperintensity of the anteroinferior capsule (outlined arrowhead). Ac, acromion; Cl, clavicle; Co, coracoid; SS, supraspinatus; Sub, subscapularis; GL, scapular glenoid; HH, humeral head.

**Figure 4 diagnostics-13-03410-f004:**
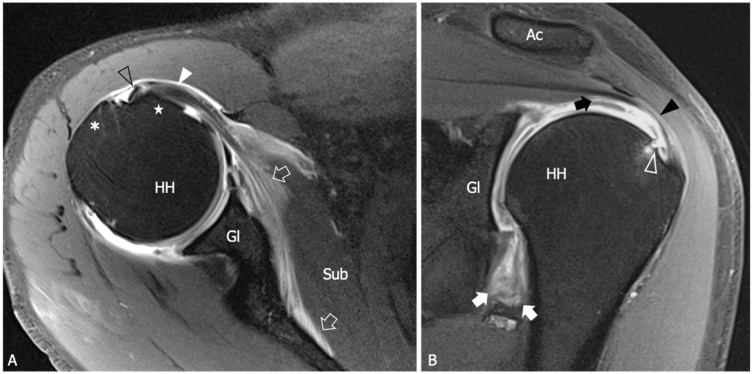
Adhesive capsulitis in a 45 year old man with severe limitation of the glenohumeral ROM after a trauma, who was submitted for an MRA for a suspected labral tear. (**A**) Axial and (**B**) Coronal tSE fat-suppressed T1-weighted MR images obtained after intraarticular injection of gadolinium demonstrate anterior extravasation of the contrast medium (outlined arrows) into and underneath the subscapularis muscle (Sub) as a consequence of capsular stiffness and fissuration. Note the abnormally low distension of the axillary recess (arrows) and its markedly thickened walls. In (**B**) a partial thickness tear (white outlined arrowhead) of the articular side of the supraspinatus tendon (black arrowhead) is also evident. In effect, the tear involves both the inner fibers of the supraspinatus and the joint capsule, which are merged at this level to form the superior complex. As a consequence, note the superior migration of the contrast outside the joint cavity (black arrow). Black outlined arrowhead, long head of the bicep tendon; white arrowhead, subscapularis tendon; asterisk greater tuberosity; star, lesser tuberosity; HH, humeral head; Gl, glenoid; Ac, acromion.

**Figure 5 diagnostics-13-03410-f005:**
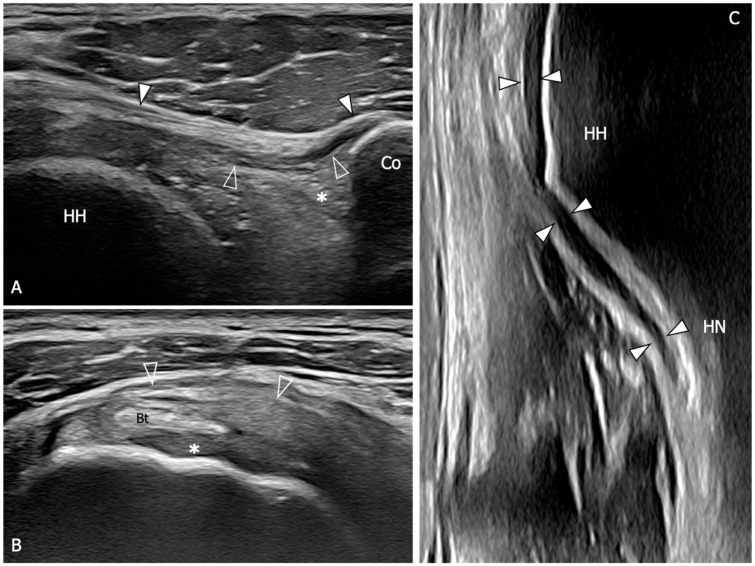
Glenohumeral joint capsule and pericapsular ligaments, normal US findings. (**A**) Oblique transverse 18–5 MHz US image shows the normal thin and fibrillar appearance of the coracohumeral ligament (outlined arrowheads), which is demonstrated connecting the coracoid (Co) and humeral head (HH) in a deeper position respective to the coracoacromial ligament (arrowheads). Note the homogeneous and hyperechoic appearance of the subcoracoid fat (asterisk). (**B**) Short-axis 18–5 MHz US image shows the distal part of the coracohumeral ligament (outlined arrowheads) in the area of the rotator interval and the biceps pulley (asterisk). Bt, long head of the biceps tendon. (**C**) Longitudinal 18–5 MHz US obtained orienting the probe parallel to the humerus in the axillary region shows the inferior capsule (arrowheads) overlying the humeral head (HH) and folding over the humeral neck (HN).

**Figure 6 diagnostics-13-03410-f006:**
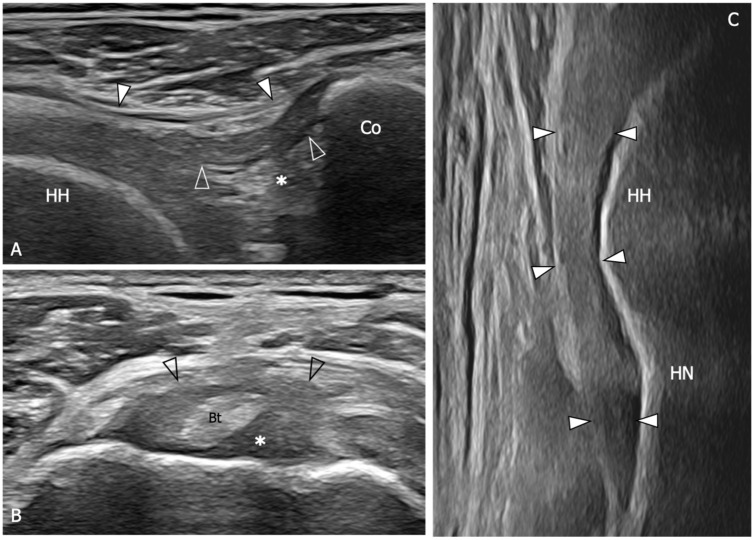
US findings in a 53 year old woman with adhesive capsulitis. (**A**) Oblique transverse 18–5 MHz US image demonstrates the markedly thickened coracohumeral ligament (outlined arrowheads), which has lost the normal fibrillar echotexture and appears homogeneously hypoechoic due to fibrotic changes and degeneration of the fibers. Note the presence of hypoechoic synovial tissue in the subcoracoid triangle (asterisk). Arrowheads, coracoacromial ligament. (**B**) Short-axis 18–5 Mhz US evidences the thickening and fibrotization of the coracohumeral ligament (arrowheads) and the biceps pulley (asterisks) in the rotator interval. (**C**) Longitudinal 18–5 MHz US image shows a significant thickening of the inferior capsule (arrowheads). HH, humeral head; Co, coracoid; Bt, long head of the biceps tendon; HN, humeral neck.

**Figure 7 diagnostics-13-03410-f007:**
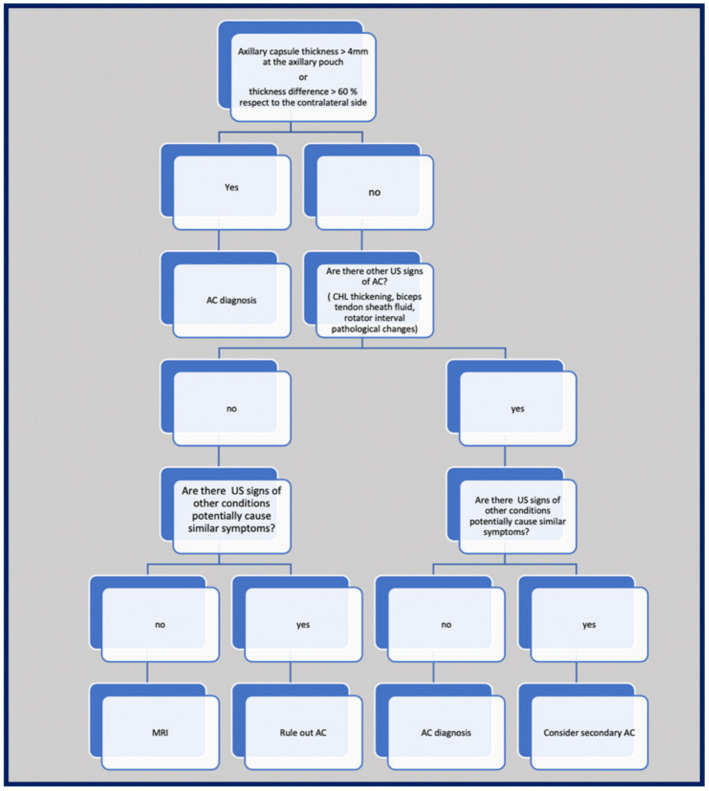
Protocol for imaging evaluation of patients with suspected AC.

**Table 1 diagnostics-13-03410-t001:** Evidence on MRI findings in AC.

	References
Thickening of the coracohumeral ligament, fat obliteration of the rotator interval, hyperintensity and thickening of the inferior glenohumeral ligament, and contrast enhancement of the axillary joint capsule and the rotator interval are the most accurate signs of AC. The sensitivity and specificity of inferior glenohumeral thickening detected on conventional MRI are not significantly different from those detected on direct MR Arthrogram: consequently, the non-arthrogram MRI is recommended for AC diagnosis.	[53]
The rotator interval capsule thickness ≥7 mm has a specificity of 86% and a sensitivity of 64% for AC diagnosis. A coracohumeral ligament thickness ≥4 mm has high specificity (95%) but lower sensitivity (59%) for AC. Obliteration of the triangular fat pad inferior to the coracohumeral ligament has high specificity (100%) and poor sensitivity (32%).	[54]
Thickening of the rotator interval over 6 mm on sagittal oblique proton-density images may correlate with the patient’s range of rotational motion. An axillary recess capsule thickness of more than 4.5 mm measured on T1 oblique coronal images demonstrated the greatest diagnostic accuracy for AC diagnosis, with a sensitivity, specificity, and overall accuracy of 91%, 90%, and 90%, respectively.	[55]
Obliteration of the subcoracoid fat triangle has been more frequently observed in early stages of AC. Capsule thickness and hyperintensity on proton density sequence correlate with clinical stages.	[56]
Hyperintensity in the axillary pouch/inferior glenohumeral ligament complex on MRI using non-arthrography T2-weighted fat-suppressed sequences demonstrated high sensitivity (85.3–88.2%) and specificity (88.2%) and low variability among different observers with a kappa value of 0.85.	[57]
An axillary recess capsule thicker than 4 mm on T1 oblique coronal MR images suggests a diagnosis of AC with a sensitivity of 70% and a specificity of 95%.	[58]
A positive linear correlation is demonstrated between the grade of axillary recess capsule enhancement, the thickness of the joint capsule, and the intensity of pain in individuals with AC. No association was observed between the aforementioned parameters and the severity of range of motion limitation.	[59]
No differences in the accuracy of AC diagnosis emerged between conventional MRI and gadolinium-enhanced MRI despite the intravenous administration of contrast agent demonstrated to have some effects in increasing the reader’s confidence in measuring the joint capsule.	[60]

**Table 2 diagnostics-13-03410-t002:** Evidence on US findings in AC.

	References
An 88% sensitivity (95%CI: 74–95) and a 96% specificity (95%CI: 88–99) are demonstrated when US detect inferior capsule and coracohumeral thickening, rotator interval abnormality, and restricted range of motion.	[10]
The mean thickness of the axillary pouch capsule in patients with AC measured with US is 4 mm versus 1.3 mm in asymptomatic controls.	[61]
The axillary capsule thickness measured at its widest portion is 4.4 mm in the affected shoulder and 2.2 mm in the unaffected shoulder (*p* < 0.001). US measurements demonstrated good correlation with MR (*p* < 0.001, r = 0.83).	[62]
A cutoff value of 4 mm for axillary pouch thickness yielded a sensitivity of 93.8% and a specificity of 98% in diagnosing AC. A difference of 60% between the affected and the unaffected side may help in disclosing this condition also in patients with suggestive symptoms but axillary recess thickness less than 4 mm.	[63]
The average CHL thickness measure both in short and long axis was 3 mm in shoulders affected by AC. Painful shoulders without AC diagnosis exhibited an average coracohumeral ligament thickness of 1.39 mm; asymptomatic shoulders had an average coracohumeral ligament thickness of 1.34 mm.	[64]
Patients with AC exhibited a significantly thicker coracohumeral ligament (1.2 mm) compared to both subjects with painful shoulders (0.54 mm) and healthy volunteers (0.4 mm).	[65]
Patients with AC demonstrated a higher prevalence of effusion in the long head of biceps tendon sheath in the affected shoulder with respect to the contralateral side.	[66]
A greater amount of the long head of the biceps sheath effusion was found in patients with AC compared to patients with other causes of a painful shoulder. A negative correlation between the amount of effusion was found within the long head of the biceps tendon sheath and the glenohumeral range of motion.	[67]
Limitation in subacromial gliding of the supraspinatus tendon is found in 70.1% of patients with AC. The limitation of the supraspinatus tendon gliding beneath the acromion was demonstrated to be inversely correlated with the maximum amount of intra-articular injection for MRA.	[68]
US evidence of hypervascular soft tissue within the rotator interval demonstrated high sensitivity (97%) and specificity (100%) for the diagnosis of AC.	[69]
Microvascular analysis of the subcoracoid triangle demonstrated a sensitivity of 76.92% and a specificity of 91.43% in diagnosing AC. The cutoff for the area of vascular flow detected during the examination of the subcoracoid triangle was set at 1.31 mm^2^.	[70]
The injected volume of sonographic contrast did not differ significantly between healthy controls and patients with AC (19.0 ± 0.22 mL vs. 18.3 ± 0.29 mL, *p* = 0.07). However, the latter had a significantly smaller axillary recess volume compared to controls and showed more frequent filling defects in the joint cavity and synovitis-like abnormalities (91.1% vs. 13.3% and 75.6% vs. 22.2%, respectively).	[71]

## Data Availability

Not applicable.

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
