# Peer review of "Adhesive Capsulitis of the Shoulder: Current Concepts on the Diagnostic Work-Up and Evidence-Based Protocol for Radiological Evaluation"

_diagnostics, 2023, doi:10.3390/diagnostics13223410_

Round 1
Reviewer 1 Report
Comments and Suggestions for Authors
I congratulate with you for this high-quality review article on a current 'hot topic' in musculoskeletal radiology: Imaging of adhesive capsulitis.
I have some request of changes/improvements in some section of this manuscript:
- 'Epidemiology and risk factors'. In this section there is a major risk factor for AC that is missing: Calcific tendinopathy. Calcific Tendinopathy has several possible complications and one of those is certainly a secondary adhesive capsulitis.
Indeed, Merolla and Porcellini in 2015 (PMID: 25697847) recognized and reported AC as a main complication of calcific tendinopathy (in a sort of expert review-paper). Moreover, recently Spinnato et al. in 2023 (PMID: 37198078) reported this association (CT complicated with AC), and proposed the possibility of a combined ultrasound-guided treatment. I suggest You to include this risk factor and refer to these 2 above mentioned articles.
In the previous decades, when CT and AC, often referred to surgery other authors mentioned this association, and proposed an arthroscopic combined treatment (PMID: 18281224)
- 'Natural History' section. I suggest You to summarize the phases of the disease into bulletin point (or into a Table).
- In the single imaging tools (MRI, US..) I suggest to summarize the main results reported, with bibliographic references in tables.
- 'Ultrasound'. Long head biceps tendon sheet fluid effusion is currently reported in several US study in association with AC, resulting in being one of the most important imaging feature for diagnosis. In this section you cited only the study by Park et al., I suggest You to remark this concept and this US sign of AC, with reference to other studies (particularly I suggest You to refer to this recent one research article published in Diagnostics (Basel): PMID: 36140631)
- 'Ultrasound' section. I suggest You to briefly discuss/cite that as a great advantage, Ultrasound may allow a direct percutaneous treatment (hydrodistensions, drug injection, neuromodulation...) simoultaneously to the diagnosis of AC (suggested ref. PMID: 34872482)
- 'Suggested imaging protocol for patients with clinically suspected AC' section. I suggest You to remove calcific tendinopathy from differential diagnosis (or confuonding factors) since as previously stated calcific tendinopathy can be complicated and result in AC.
- 'Suggested imaging protocol for patients with clinically suspected AC'. In this section you correctly try to summarize the most important imaging features related to AC. I agree with you that undoubtedly axillary pouch thickening (>4mm or greater than 60% of contralateral shoulder) is the main feature that allows the diagnosis of AC. Beneath this feature I think that as you correctly stated there are other 'minor' factors/features such as RI thickeness or CH ligament thickening alone; all these factors differing in cut-off values of normality depending on type of measurements or other bias. Due to that I agree that these features should be considered as secondary ones, and used in doubtful cases/diagnoses. On the contrary, the presence/absence of fluid in the long biceps tendon sheet, is a certain dichotomous feature, supported by several research article in association with AC (PMID: 34940958 - PMID: 36140631 - PMID: 33997079). This feature, is relevant, simple to assess and do not need PD evaluation. I suggest You to include this feature after Axillary pouch thickening as an important element for this diagnosis.
- Finally, I suggest You to summarize these key-element for the diagnosis in a table or in a graphic figure.
Author Response
Thank you for the very accurate e thorough revision. We have considered all the suggestions and tried to implement them in our manuscript.
- We have included the calcific tendinitis among the risk factors of AC and cited the papers proposed.
- The natural history of AC have been summarized as bulletin points.
- The main results of our literature research regarding the different imaging diagnostic approaches to AC have been summarized in two tables, one for Ultrasound and one for MRI.
- Discussion regarding the long head biceps tendon sheath fluid effusion as a feature of AC has been extended.
- The use of the Ultrasound to guide interventional procedure has been mentioned in the paragraph " Role of imaging"
- Regarding the suggestion of removing the Calcific tendinitis from the alternative diagnosis respect to AC, we think that the two conditions have different therapeutic options and, although they may coexist, it is clinically relevant to distinguish the two syndromes often presenting with overlapping symptoms.
- In our diagnostic protocol, the fluid distension of synovial sheath of the long head of the biceps has been reported as an " associated sign" of AC, since other gleno-humeral joint pathologies may lead to the fluid accumulation in the tenosynovial sheath and consequently it should not be considered as a stand-alone sign of AC.
- The key elements of the diagnostic work-up of AC have been summarized in a graphic
Reviewer 2 Report
Comments and Suggestions for Authors
The authors are to be congratulated for tackling a difficult topic - the diagnostic approach to adhesive capsulitis. The authors have considerable expertise in ultrasound/imaging, but the lack of clinical shoulder specialists on the authorship panel is reflected in the paper. As a rheumatologist with limited training in musculoskeletal ultrasound my comments focus on how I would read this as a clinician eager to use a reliable diagnostic aid for this condition. Unfortunately, at the end of the article I am not sure that the evidence they have presented is enough to change my practice. I believe that a stronger case would be made if the evidence was better presented & practical issues taken into consideration in the protocol (e.g. where severe pain and restricted ROM, MRI preferred over US). Suggestions for further research would also be helpful.

Minor improvements required
Author Response
Thank you for your observations and suggestions. The work was aimed to identify the most effective way to exploit the imaging tools in supporting and aiding the clinical diagnosis of AC, since from the clinical point of view the signs and symptoms of different conditions affecting the shoulder may overlap. Although strong evidences supporting a systematic adoption of any imaging methods for the diagnosis of Adhesive Capsulitis are still missing, the amount of literature suggesting the inclusion of Ultrasound in the diagnostic work-up of this disabling condition should not be ignored. The widely recognized as the most sensitive sign of adhesive capsulitis, the increased thickness of capsule of the axillary recess, can be easily assessed in almost all the patients even by a not-ultrasound expert with a relatively short training. Furthermore, ultrasound can also promptly rule out many other differential diagnosis or identify coexisting conditions (as calcific tendinitis, subacromial-subdeltoid bursitis, acromion-clavicular joint arthritis).
As suggested, we have included some practical points and future perspectives.
Reviewer 3 Report
Comments and Suggestions for Authors
Overall, I think this is a well-written review.
And I think the strongest point of this review is that it contains detailed descriptions and explanations from a radiology perspective.
In that respect, it delivers meaningful information to readers.
Nevertheless, several areas that need to be corrected were identified.
Title
: I think the title of this article should be revised to focus more on the radiology aspect.
Abstract
: The content needs to be revised to include sentences that convey more information.
Line45-47:this sentence seems good. it will be origin for modified title.
Line 49-64: There is a need to add more references to support this in more sentences.
Was Figure 1 created by the authors? Or is this from another paper? Please confirm.
If possible, I would like to recommend consistently expressing it as "AC" rather than frozen shoulder in this article.
In image modality, it seems appropriate for US to come first, followed by MRI.
line 246-247: "yo" should be modified as years old.
line 422-454: More review of cost effectiveness should be added. Only then will it be possible to mention this in the conclusion.
Comments on the Quality of English Language
When modifying English expressions and using phrases, parts that need to be written in accordance with the principles of academic description are identified.
Author Response
Title: I think the title of this article should be revised to focus more on the radiology aspect.
Dear Reviewer, thank you for your suggestion. Despite the focus of our work is on the radiological aspects of adhesive capsulitis, we decided to address this topic also from a clinical perspective, and this is the reason why we decided for a more inclusive title. In any case, radiology is mentioned in the second part of the title in order to draw the readers’ attention on the main topic of this work. Following your suggestion, we modified the title in Adhesive capsulitis of the shoulder: current concepts on diagnosis and evidence-based protocol for radiological evaluation.
Abstract: The content needs to be revised to include sentences that convey more information.
Dear Reviewer, thank you for your suggestion. Following this comment, we modified the abstract to include some relevant information on diagnostic findings of adhesive capsulitis, which are then discussed in the main text.
Line45-47: this sentence seems good. it will be origin for modified title.
Thank you for your suggestion.
Line 49-64: There is a need to add more references to support this in more sentences.
Dear Reviewer, thank you for this comment. Following your suggestion references 6 and 12 have been added to the manuscript.
Was Figure 1 created by the authors? Or is this from another paper? Please confirm.
Dear Reviewer, thank you for this question. Figure 1 includes an anatomic dissection and a schematic drawing. Both the images have been created by the authors.
If possible, I would like to recommend consistently expressing it as "AC" rather than frozen shoulder in this article.
Dear Reviewer, thank you for your comment. We modified the term frozen shoulder with AC when appropriate.
In image modality, it seems appropriate for US to come first, followed by MRI.
Dear Reviewer, in this work MRI has been discussed before US as until now it has been referred to as the gold standard modality for the diagnosis of adhesive capsulitis.
line 246-247: "yo" should be modified as years old.
Thank you for your suggestion. We modified the text accordingly.
line 422-454: More review of cost effectiveness should be added. Only then will it be possible to mention this in the conclusion.
Dear Reviewer, thank you for your comment. As mentioned, there is currently a lack of evidence on the cost effectiveness of imaging evaluation of patients with adhesive capsulitis. Further research is needed to investigate this parameter.
Reviewer 4 Report
Comments and Suggestions for Authors
Review
Title:” Adhesive capsulitis of the shoulder: current concepts and evidence-based protocol for radiological evaluation and diagnosis”
This study is a review article about Adhesive capsulitis of the shoulder: current concepts and evidence-based protocol for radiological evaluation and diagnosis.
It is well described article with proper method and conclusion. It is acceptable
Title
Good
Abstract
Good
Introduction
Good.
Relevant anatomy
Good.
Epidemiology and risk factors
Good.
Pathogenesis
Good.
Natural history
Good.
Clinical diagnostic workup
Good.
Role of imaging
Good.
Magnetic Resonance Imaging
Good.
Ultrasound
Good.
Suggested imaging protocol for patients with clinically suspected AC
Good.
Conclusion
Good.
References
Good.
Author Response
Title:” Adhesive capsulitis of the shoulder: current concepts and evidence-based protocol for radiological evaluation and diagnosis”
This study is a review article about Adhesive capsulitis of the shoulder: current concepts and evidence-based protocol for radiological evaluation and diagnosis.
It is well described article with proper method and conclusion. It is acceptable
Dear Reviewer, thank you for this comment. We are glad you have appreciated our work.
Round 2
Reviewer 1 Report
Comments and Suggestions for Authors
Thank You for providing all the requested changes and improvements.
Congratulations for your exhaustive review article on an important topic nowadays.